# The role of GABA in semantic memory and its neuroplasticity

JeYoung Jung[1]*, Steve Williams[2], Matthew A Lambon Ralph[3]*

[1]School of Psychology, University of Nottingham, Nottingham, United Kingdom; [2]Division of Informatics, Imaging and Data Science, School of Health Sciences, University of Manchester, Manchester, United Kingdom; [3]MRC Cognition and Brain Sciences Unit (CBU), University of Cambridge, Cambridge, United Kingdom

**\*For correspondence:**
jeyoung.jung@nottingham.ac.uk (JYJ);
Matt.Lambon-Ralph@mrc-cbu.cam.ac.uk (MALR)

**Competing interest:** The authors declare that no competing interests exist.

## eLife Assessment

Jung et al. present **valuable** work on the relationship between gamma-aminobutyric acid (GABA) levels within the anterior temporal lobes (ATL) to semantic memory while accounting for inter-individual differences. They provide **solid** evidence suggesting that inhibitory continuous theta burst transcranial magnetic stimulation (cTBS TMS) increased GABA concentration and decreased the blood-oxygen dependent signal (BOLD) during a semantic task. The results will be of interest to researchers studying the neurobiology of semantic cognition.

## Abstract

A fundamental challenge in neuroscience is understanding neural functioning and plasticity of the brain. The anterior temporal lobe (ATL) is a hub for semantic memory, which generates coherent conceptual representations. GABAergic inhibition plays a crucial role in shaping human cognition and plasticity, but it is unclear how this inhibition relates to human semantic memory and its plasticity. Here, we employed a combination of continuous theta burst stimulation (cTBS), MR spectroscopy and fMRI to investigate the role of GABA in semantic memory and its neuroplasticity. We found that inhibitory cTBS increased GABA concentrations in the ATL and reduced blood-oxygen level-dependent (BOLD) activation during semantic tasks. Crucially, changes in GABA were tightly linked to changes in regional activity, suggesting that GABA mediates cTBS-induced plasticity. Individuals with better semantic performance exhibited selective activity in the ATL, attributable to higher GABA levels, which can sharpen distributed semantic representations. Our results revealed a non-linear, inverted-U-shape relationship between GABA levels in the ATL and semantic performance, thus offering an explanation for the individual differences in semantic memory function and neuromodulation outcomes. These findings offer a neurochemical explanation for individual variability in neuromodulation and provide insights for developing targeted interventions for semantic impairments.

## Introduction

Understanding how the brain functions to drive flexible human behaviour has been a fundamental challenge in cognitive neuroscience. The ability to (re)shape our behaviours based on our experiences relies on a flexible mechanism in the brain, which is achieved through the regulation of neural excitation and inhibition (*Tatti et al., 2017*). In particular, an imbalance between excitatory and inhibitory processes has been associated with various cognitive impairments in several psychiatric disorders (*Sohal and Rubenstein, 2019*) such as autism spectrum disorder (*Horder et al., 2018*) and schizophrenia (*Reddy-Thootkur et al., 2022*). Of particular interest is the role of the neurotransmitters, gamma-aminobutyric acid (GABA) and glutamate in coordinating neural functions supporting

performance in various cognitive domains. GABA is the primary inhibitory neurotransmitter in the brain, while glutamate is the primary excitatory neurotransmitter. The balance between GABAergic and glutamatergic neurotransmission is crucial for the proper functioning of the brain and the maintenance of optimal behavioural responses in both healthy and diseased states (*Duncan et al., 2014*; *Heaney and Kinney, 2016*; *Bojesen et al., 2021*). While GABA and glutamate are associated with various cognitive processing, the mechanistic link from the neurotransmitters to human cognition is not well understood.

Research has shown that GABAergic inhibition plays a crucial role in various processes, such as sensory processing, attention, memory, and learning (*Duncan et al., 2014*; *Heaney and Kinney, 2016*; *Jung et al., 2022*). Furthermore, GABAergic neurotransmission can regulate synaptic plasticity, leading to long-term potentiation (LTP) and long-term depression (LTD) by modulating the activity of excitatory neurons (*Isaacson and Scanziani, 2011*; *Schmidt-Wilcke et al., 2018*; *Wolff et al., 1993*). Understanding GABAergic inhibition is crucial in uncovering the neurochemical mechanisms underlying human cognition and its neuroplasticity. Research has revealed a link between variability in the levels of GABA in the human brain and individual differences in cognitive behaviour (for a reveiw, see *Duncan et al., 2014*). Specifically, GABA levels in the sensorimotor cortex were found to predict individual performance in the related tasks: higher GABA levels were correlated with a slower reaction time in simple motor tasks (*Stagg et al., 2011a*) as well as improved motor control (*Boy et al., 2010*) and sensory discrimination (*Puts et al., 2011*; *Kolasinski et al., 2017*). Visual cortex GABA concentrations were positively correlated with a stronger orientation illusion (*Song et al., 2017*), a prolonged binocular rivalry (*Pitchaimuthu et al., 2017*), while displaying a negative correlation with motion suppression (*Pitchaimuthu et al., 2017*). Individuals with greater frontal GABA concentrations demonstrated enhanced working memory capacity (*Yoon et al., 2016*; *Porges et al., 2017*). Studies on learning have reported the importance of GABAergic changes in the motor cortex for motor and perceptual learning: individuals showing bigger decreases in local GABA concentration can facilitate this plasticity more effectively (*Stagg et al., 2011a*; *Floyer-Lea et al., 2006*; *Kolasinski et al., 2019*; *Lea-Carnall et al., 2020*). However, the relationship between GABAergic inhibition and higher cognition in humans remains unclear. The aim of the study was to investigate the role of GABA in relation to human higher cognition – semantic memory and its neuroplasticity at the individual level.

Semantic memory is a crucial aspect of human cognition, encompassing our knowledge of concepts and meaning, such as words, people, objects, and emotion (*Hodges et al., 2000*; *Lambon Ralph, 2014*). Accumulating and converging evidence indicates that the anterior temporal lobe (ATL) is a transmodal and transtemporal hub of semantic memory that generates coherent semantic representations through interactions with modality-specific brain regions and integration over time/episodes (*Lambon Ralph, 2014*; *Ralph et al., 2017*; *Patterson et al., 2007*). The initial and strong evidence supporting this hypothesis comes from semantic dementia patients who show selective semantic degradation in both verbal and non-verbal domains due to progressive ATL-centred atrophy (*Hodges et al., 2000*; *Bozeat et al., 2000*; *Hodges and Patterson, 2007*; *Ralph and Patterson, 2008*). Recent studies have also supported this hypothesis using intracranial recordings and cortical stimulation (*Chen et al., 2016*; *Shimotake et al., 2015*), magnetoencephalography (*Clarke et al., 2011*; *Mollo et al., 2017*) and functional magnetic resonance imaging (fMRI; *Coutanche and Thompson-Schill, 2015*; *Murphy et al., 2017*; *Peelen and Caramazza, 2012*; *Visser et al., 2012*). Transcranial magnetic stimulation (TMS) studies have further established the link between ATL and semantic memory. Perturbing the ATL with inhibitory repetitive TMS (rTMS) and theta burst stimulation (TBS) resulted in healthy individuals showing slower reaction time during semantic processing (*Jung and Lambon Ralph, 2016*; *Pobric et al., 2007*; *Binney et al., 2010*; *Pobric et al., 2010*, *Lambon Ralph et al., 2009*; *Jung and Lambon Ralph, 2021*; *Pobric et al., 2009*). Our investigations combining rTMS/TBS with fMRI have revealed the critical role of the ATL in the neuroplasticity of the semantic system, demonstrating the flexible and adaptive nature of the neural mechanisms underpinning semantic memory function (*Jung and Lambon Ralph, 2016*; *Binney and Ralph, 2015*; *Jung et al., 2021*). Despite the compelling and consistent findings regarding the involvement of the ATL in semantic memory and its capacity for neuroplasticity, the specific ways in which the underlying neurotransmitter systems influence ATL function in semantic memory and its neuroplasticity remain unclear.

Previously, we explored neurotransmitter systems on the functioning of the ATL in semantic memory using a combination of magnetic resonance spectroscopy (MRS) and fMRI (*Jung et al.,*

*2017*). By utilising MRS, a non-invasive method for measuring neurometabolites such as GABA and glutamate in vivo, we were able to detect and quantify regional GABA and glutamate concentrations in the ATL (*Sanaei Nezhad et al., 2018*). The concentration of GABA in the ATL showed a positive correlation with performance in semantic tasks and was negatively associated with blood-oxygen level-dependent (BOLD) signal changes during semantic processing. Our results highlighted the critical involvement of GABAergic inhibition in the modulation of neural activity and behaviour related to semantic processing in the ATL. Subsequently, we explored the relationship between regional GABA levels in the ATL and cTBS-induced plasticity in semantic memory (*Jung et al., 2022*). To achieve this, we acquired the ATL MRS and fMRI prior to stimulation and then delivered cTBS, an inhibitory protocol (*Huang et al., 2005*), to the ATL. We examined how baseline GABA levels in the ATL were associated with changes in semantic task performance after cTBS. The results showed that individuals with higher GABA levels in the ATL exhibited stronger cTBS effects on semantic processing, especially those who displayed inhibitory responses after cTBS. These findings suggest that the GABAergic action in the ATL plays a crucial role in cTBS-induced plasticity in semantic memory, predicting inter-individual variability of cTBS responsiveness.

Based on these findings, we hypothesised that GABAergic inhibition in the ATL can affect neural dynamics within the ATL underpinning semantic memory and its neuroplasticity. The study investigated the neural mechanisms underlying cTBS-induced neuroplasticity in the ATL by linking neurochemical profiles, task-induced regional activity, and individual variability in semantic memory performance. We used a combination of cTBS, MRS and fMRI to examine the relationship between changes in GABA levels, cortical activity during a semantic task, and semantic task performance. First, we hypothesised that the inhibitory cTBS would increase GABA concentrations and decrease task-induced BOLD signal changes in the ATL. Second, the effects of cTBS on semantic processing can be attributed to GABAergic action in the ATL at the individual level: greater changes in GABA concentrations would result in increasing changes in task-induced BOLD signal during semantic processing and semantic task performance. Furthermore, to address and explore the relationship between regional GABA levels in the ATL and semantic memory function, we combined data from our previous study (*Jung et al., 2017*) with the current study's data. We then explored the function of GABA in the ATL in relation to semantic function. Finally, we extended this GABAergic function to semantic neuroplasticity driven by cTBS.

## Results

We acquired resting-state MRS for the left ATL and vertex followed by fMRI before and after cTBS (*Figure 1A*). During fMRI, participants made semantic association decisions as an active task and pattern matching as a control task (*Figure 1B*). GABA concentrations were estimated from the ATL with the vertex as a control region (*Figure 1C*). cTBS with 80% of resting motor threshold (RMT) was delivered outside of scanner at one of the target regions with a week gap between two sessions (*Figure 1D*).

### cTBS modulates regional GABA concentrations and task-related BOLD signal changes in the ATL

E-field modelling of cTBS showed that, as intended, ATL cTBS stimulated the left ventrolateral ATL (*Figure 2A*). To investigate how cTBS modulates GABA concentrations, we quantified GABA/NAA and calculated the changes (POST− PRE). A 2×2 repeated measures analysis of variance (ANOVA) with stimulation (ATL vs. vertex) and VOI (ATL vs. vertex) as within subject factor was performed. There was a significant interaction effect between the stimulation and VOI ($F_{1,16}$ = 4.57, p=0.048; *Figure 2B*). There was no significant main effect of the stimulation ($F_{1,16}$ = 3.23, p=0.091) and VOI ($F_{1,16}$ = 0.64, p=0.435). Planned paired t-tests revealed that ATL stimulation significantly increased GABA concentrations in the ATL compared to the control stimulation (t=1.86, p=0.040) and control site (t=2.07, p=0.027). There were no cTBS effects in the vertex VOI regardless of the stimulation (ps >0.23). It is noted that there was no significant cTBS effect in Glx (*Figure 2—figure supplement 1*).

fMRI results demonstrated that the semantic association task evoked increased activation in the ATL, prefrontal and posterior temporal cortex compared to the control task (*Figure 2C*, *Figure 2—figure supplement 2* and *Supplementary file 1*). To examine the effects of cTBS, we performed ROI

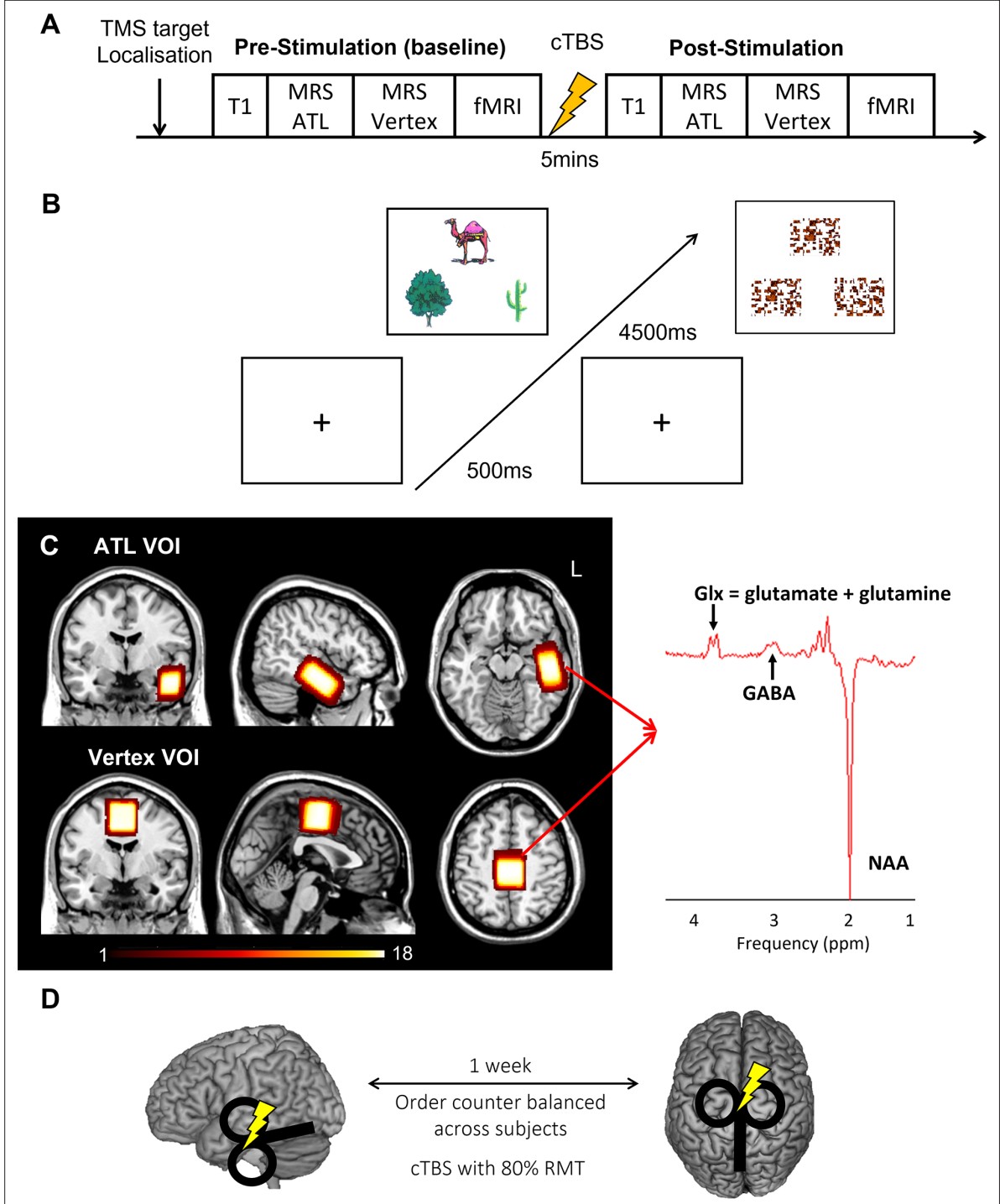

**Figure 1.** Experimental design and procedure. (**A**) Experimental procedure. (**B**) An example of the semantic association task (left) and control task (right: pattern matching). Each trial starts with a fixation followed by stimuli, which have three items, a target on the top and two choices at the bottom. (**C**) The location of volume of interest (VOI) for MRS (left ATL and vertex) and a representative MRS spectrum with estimated peaks (right). Colour bar indicates the number of overlapping participants. NAA: N-acetylaspartate. (**D**) cTBS protocols. cTBS was applied over the left ATL and vertex as a control site. Each stimulation was delivered on different days with a week gap at least.

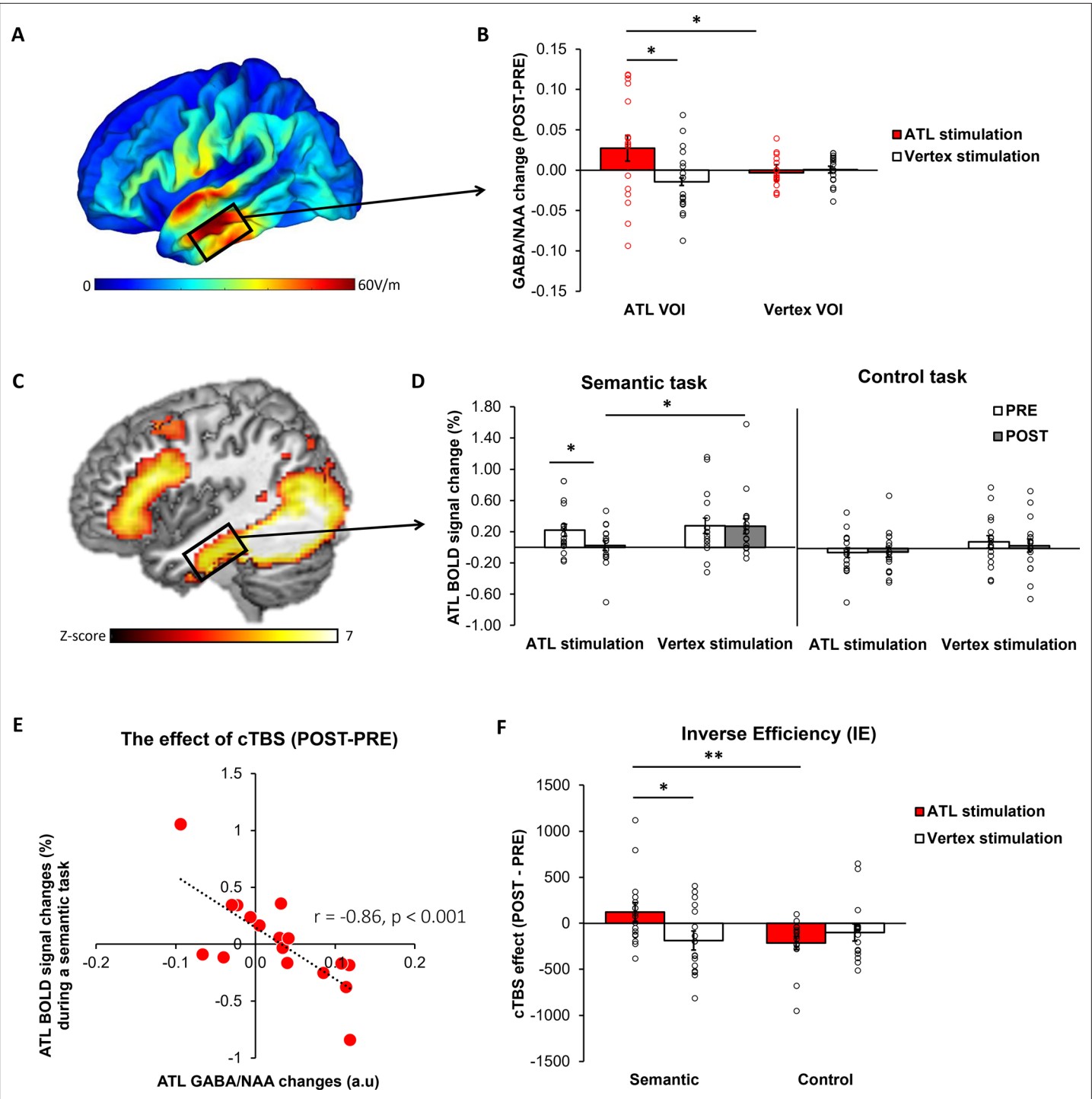

**Figure 2.** The effects of cTBS in the ATL. (**A**) ATL cTBS e-field modelling. (**B**) cTBS-induced regional GABA changes in the ATL. Red bar indicates the ATL stimulation and white bar indicates the control (vertex) stimulation. (**C**) fMRI results of the contrast of interest (semantic >control) in the ATL pre-stimulation session. (**D**) cTBS-induced ATL BOLD signal changes during a semantic and control task. White bars represent the pre-stimulation session, and grey bars represent the post-stimulation session. (**E**) The relationship between cTBS-induced GABA changes and BOLD signal changes in the ATL. (**F**) The results of task performance. A positive value of cTBS effect (Post – Pre) in IE suggests an inhibitory effect, indicating poorer performance after the stimulation. In contrast, a negative value denotes a facilitatory effect, signifying improved performance following the stimulation. Red bar indicates the ATL stimulation and white bar indicates the control (vertex) stimulation. Each individual is represented as a circle. * p<0.05, ** p<0.01.

The online version of this article includes the following figure supplement(s) for figure 2:

**Figure supplement 1.** The effects of cTBS in Glx.

*Figure 2 continued on next page*

**Figure supplement 2.** The results of fMRI (semantic >control).

**Figure supplement 3.** The results of task performance.

**Figure supplement 4.** E-field modelling of vertex stimulation.

**Figure supplement 5.** The location of the ATL target site (MNI −36,−15, −30) and its regional activity during semantic processing.

analysis using the same VOI in the ATL. Planned paired t-tests revealed that BOLD signal changes during semantic processing were significantly altered after ATL cTBS compared to the pre-stimulation (t=1.78, p=0.046) and the control stimulation (t=−2.11, p=0.025; *Figure 2D*).

To investigate the effects of ATL cTBS, we conducted a partial correlation analysis between GABA changes (POST–PRE) and BOLD signal changes (POST– PRE), accounting for age and sex. We found a significant correlation between cTBS-induced GABA changes and BOLD signal changes in the ATL (*r*=−0.86, p<0.001). Individuals with greater increases in ATL GABA levels following ATL cTBS showed greater reductions in task-induced BOLD signal changes in the ATL. These results demonstrate that ATL cTBS modifies regional GABA concentrations, and the cTBS-induced changes in GABA levels are connected to individual-level changes in task-related fMRI signal.

## cTBS disrupts semantic task performance, revealing substantial individual variability in responsiveness

Participants' performance was examined using a 2×2 repeated measures ANOVA with stimulation (ATL vs. vertex) and session (PRE vs. POST) as within-subject factors. There were no significant main effects and interactions on reaction time (RT) in the semantic task (Fs >0.19, ps >0.220). However, we found a significant main effect of session in the control task ($F_{1, 15}$ = 20.21, p<0.001). *Post hoc* paired t-tests demonstrated that participants performed the task faster in the post-session compared to the pre-session, except in the semantic task after the ATL stimulation. The results showed that ATL cTBS attenuated the practice effects found in the control stimulation and control task. There were no significant effects in accuracy (Fs >0.01, ps >0.073). The results of accuracy and RT for each task were summarised in the *Supplementary file 1* and *Figure 2—figure supplement 3*.

To evaluate the cTBS effects in behaviour, we used the inverse efficiency (IE) score (RT/1-the proportion of error) and calculated IE changes (POST-PRE). The cTBS effects on participants' performance were examined using a 2×2 repeated measures ANOVA with stimulation (ATL vs. vertex) and task (semantic vs. control) as within-subject factors. There was no significant main effect of stimulation. However, we found a significant main effect of task ($F_{1, 15}$ = 6.66, p=0.021) and a marginally significant interaction between the stimulation and task ($F_{1, 15}$ = 4.06, p=0.061). *Post hoc* paired t-tests demonstrated that participants performed the semantic task worse (higher IE score) after the ATL stimulation compared to the control task (t=2.81, p=0.006) and vertex stimulation (t=1.91, p=0.038; *Figure 2F*). The results showed that ATL cTBS induced the task-specific inhibitory effects on semantic task performance. It is noted that a higher IE score indicates poorer performance.

Moreover, we categorised participants based on changes in their semantic task performance following ATL stimulation to examine the relationship between ATL GABA levels and individual behavioural response to cTBS. To account for practice effects (*Supplementary file 1*), we first adjusted task performance (IE) by normalising ATL stimulation performance relative to with vertex stimulation performance. A 2×2 × 2 ANOVA was conducted with task (semantic vs. control) and session (PRE vs. POST) as within-subject factors, and group (responders vs. non-responders) as a between-subject factor. The analysis revealed a significant interaction between the session and group ($F_{1, 15}$ = 10.367, p=0.006), a marginally significant interaction between the session and task ($F_{1, 15}$ = 4.370, p=0.054), and a significant three-way interaction between the session, task, and group ($F_{1, 15}$ = 7.580, p=0.015). Post hoc tests showed that after ATL stimulation, responders exhibited poorer semantic task performance (higher IE; t=−5.281, p<0.001), whereas non-responders demonstrated paradoxical, facilitatory effects on semantic task performance (lower IE; t=3.206, p=0.007; *Figure 3A*). Additionally, responders showed poorer semantic task performance compared to non-responders after ATL stimulation (t=2.349, p=0.033; *Figure 3A*). Notably, no differences were observed between responders and non-responders in the control task performance across pre- and post-stimulation sessions, confirming that the practice effect was successfully controlled (*Figure 3B*).

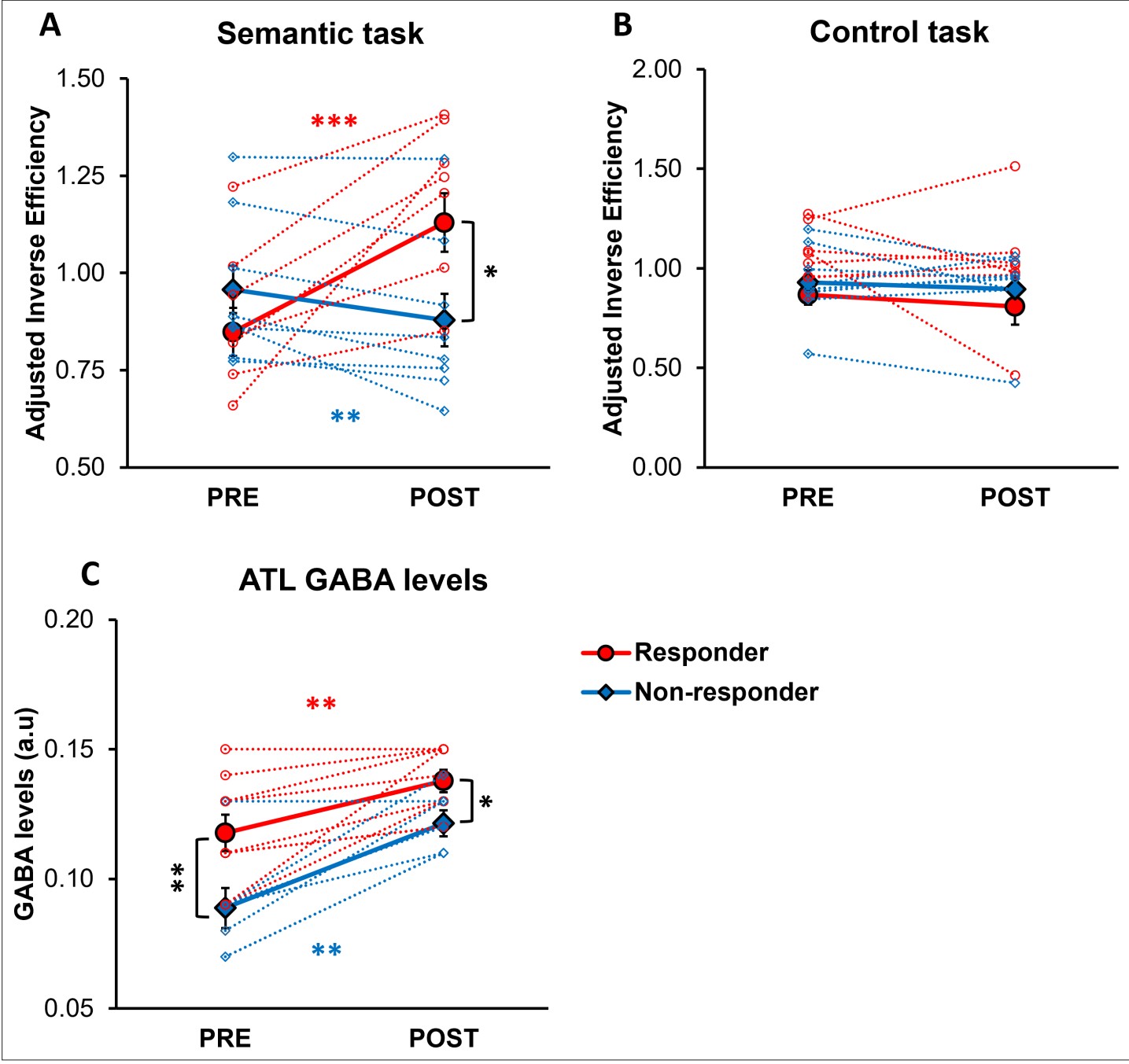

**Figure 3.** The effects of cTBS on behaviour and individual variability in responsiveness to cTBS. (**A**) Semantic task performance (IE) in pre- and post-ATL stimulation session. (**B**) Control task performance (IE) in pre- and post-ATL stimulation session. To account for the practice effect, task performance was adjusted by dividing ATL stimulation performance by vertex stimulation performance. (**C**) ATL GABA levels in pre- and post-ATL stimulation session. The red circle represents the responder, while the blue diamond denotes the non-responder. Each individual is represented as a circle. Error bars indicate standard errors. * p<0.05, ** p<0.01, *** p<0.001.

A 2x2 ANOVA with session (PRE vs. POST) as a within- subject factor and with group (responders vs. non-responders) as a between-subject factor was conducted to investigate the effects of individual cTBS responsiveness on ATL GABA levels. The analysis revealed a significant main effect of session ($F_{1, 14} = 39.906$, p<0.001) and group ($F_{1, 14} = 9.677$, p=0.008). *Post hoc* paired t-tests revealed that both responders and non-responders showed increased GABA levels in the ATL following stimulation (responder: t=–3.885, p=0.002, non-responder: t=–4.831, p=0.001; *Figure 3C*). *Post hoc* t-tests

further revealed a significant difference in ATL GABA levels between responders and non-responders in both pre-stimulation (t=2.816, p=0.007) and post-stimulation session (t=2.555, p=0.011; *Figure 3C*).

## Regional GABA concentrations in the ATL play a crucial role in semantic memory

In our prior study (*Jung et al., 2017*), ATL GABA levels were significantly and negatively correlated with ATL activity during semantic processing (*Figure 4A*). Here, we replicated our previous findings in a different cohort with the same research paradigm (pre-stimulation session). We conducted a single-voxel regression analysis with the individual's GABA concentrations (ATL pre-stimulation session) as the regressor of the fMRI contrast of interest (semantic >control). The BOLD response in the ventral ATL was significantly and negatively correlated with the individual GABA levels in the ATL (MNI –42–6 –33, p $_{SVC-FWE}$ <0.05), overlapping with the results from our previous study (*Jung et al., 2017*; *Figure 4A*). The GABA-related region of the ventral ATL overlapped with the semantic coding hotspot from electrocorticograms (ECoG) data and direct cortical stimulation (*Chen et al., 2016*; *Shimotake et al., 2015*; *Figure 4A*). Furthermore, we found that individual GABA concentrations in the ATL were positively associated with semantic task performance (*Figure 4B*). We also confirmed this finding, demonstrating that individuals with more GABA in the ATL performed the semantic task better (higher accuracy; r=0.50, p=0.035; *Figure 4B*). It should be noted that individual GABA levels also significantly correlated with ATL activity and semantic task performance at the vertex stimulation session (*Figure 4—figure supplement 1*). There was no significant relationship between ATL GABA levels and RT during semantic processing (ps >0.44; *Figure 4—figure supplement 2*) and between ATL GABA levels and control task performance (*Supplementary file 1*). These results demonstrate that higher levels of cortical GABA in the ATL are associated with task-related regional activity as well as enhanced semantic function.

## The inverted U-shaped function of ATL GABA concentrations in semantic processing

The pattern of correlation between GABAergic activity and semantic task accuracy observed in our previous study was replicated in an entirely new cohort in the current study. Next, we combined the two studies (N=37) in order to fully investigate the potential role of GABA in the ATL as a mechanistic link between ATL inhibitory GABAergic action and semantic task performance (accuracy). First, we tested the linear relationship between ATL GABA levels and semantic task performance. We confirmed our previous findings that individuals with higher GABA levels in the ATL showed better semantic task performance ($R^2$=0.49, p<0.001; *Figure 4C*). Second, to test our hypothesis, we assumed that semantic performance follows an inverted U-shaped (quadratic) function with relation to ATL GABA concentrations. In other words, people who have low or excessive GABA levels in the ATL perform the semantic task relatively poorly. The results revealed that the inverted U-shaped function between ATL GABA and semantic performance was significant ($R^2$=0.67, p<0.001; *Figure 4C*). To compare two different models, we calculated the Bayesian Information Criterion (BIC) as a measure of model fitness (*Vrieze, 2012*) and performed a partial F-test to determine whether there is a statistically significant difference between the two models. A best model fitness can be characterised by low BIC and high $R^2$. The results showed a BIC value of 243.72 for the linear function and a value of 233.36 for the quadratic function. The results of F-tests revealed that the inverted U-shaped model provided a statistically significantly better fit than the linear model (F=15.60, p<0.001). The best-fitting model is therefore the inverted-U-shaped function of ATL GABA in semantic processing. There was no significant relationship between ATL GABA levels and RT during semantic processing (linear function $R^2$=0.21, p=0.45, quadratic function: $R^2$=0.17, p=0.21).

We performed the same analysis on the pre- and post-stimulation data in order to investigate the role of ATL GABA in semantic plasticity. We found that there was a significant linear relationship between the ATL GABA levels and semantic performance before and after stimulation ($R^2$=0.19, p<0.01; *Figure 4D*). The inverted U-shaped function also showed a significant association between them ($R^2$=0.33, p<0.005; *Figure 4D*). The F-test demonstrated that the quadratic model showed a significantly better fit than the linear model (F=6.64, p=0.014). The inverted U-shaped function has the better BIC score for explaining changes in ATL GABA levels and semantic performance induced

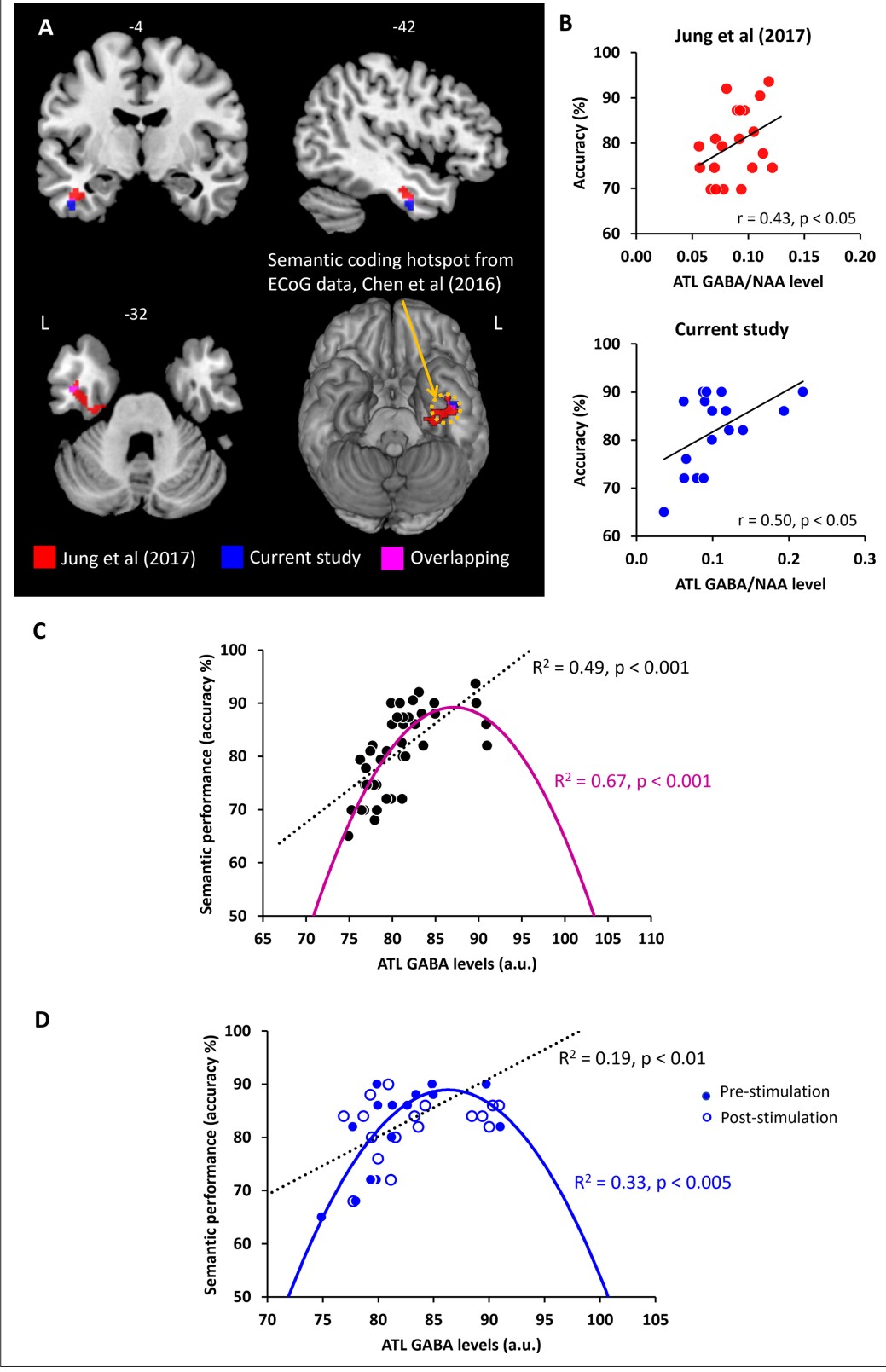

**Figure 4.** The relationship between ATL GABA levels and semantic memory. (**A**) Local maxima of the voxel-wise regression analysis of the contrast (semantic >control) with GABA concentrations in the ATL. (**B**) The relationship between individual GABA levels in the ATL and semantic task performance from our previous study (***Jung et al., 2017***) and current study (pre-stimulation session). (**C**) The ATL GABA function in relation to semantic performance.

*Figure 4 continued on next page*

*Figure 4 continued*

(**D**) The relationship between cTBS-induced changes in ATL GABA levels and semantic task performance. Dotted line represents the linear function between ATL GABA levels and semantic task performance. Coloured line represents the inverted U-shaped (quadratic) function between ATL GABA levels and semantic task performance.

The online version of this article includes the following figure supplement(s) for figure 4:

**Figure supplement 1.** The relationship between the ATL GABA levels and semantic task performance in the vertex stimulation.

**Figure supplement 2.** The relationship between the ATL GABA levels and semantic task performance (RT).

by cTBS (linear model BIC 230.21, quadratic model BIC 227.13). Thus, the best-fitting model is the inverted U-shaped for the ATL GABA changes induced by cTBS in relation to semantic function.

## Discussion

We investigated the role of cortical GABA in the ATL on semantic memory and its neuroplasticity. Our results demonstrated an increase in regional GABA levels following inhibitory cTBS in human associative cortex, specifically in the ATL, a representational semantic hub. Notably, the observed increase was specific to the ATL and semantic processing, as it was not observed in the control region (vertex) and not associated with control processing (visuospatial processing). Our study also found that the magnitude of cTBS-modulated GABA changes at the individual level was associated with their changes in ATL activity during semantic processing. Furthermore, our data confirmed and replicated our previous findings that GABA concentrations in the ATL shape task-related cortical activity and semantic task performance. In other words, individuals with greater semantic performance exhibit selective activity in the ATL due to higher concentrations of inhibitory GABA. GABAergic inhibition could sharpen activated distributed semantic representations through lateral inhibition, leading to improved semantic acuity (*Jung et al., 2017*), which aligns with theories on representational sharpening in visual perception (*Desimone, 1996*; *Kok et al., 2012*). Importantly, our data revealed, for the first time, a non-linear, inverted-U-shape relationship between GABA levels in the ATL and semantic function, by explaining individual differences in semantic task performance and cTBS responsiveness. Understanding the link between neurochemistry and semantic memory is an important step in understanding individual differences in semantic behaviour and could guide therapeutic interventions to restore semantic abilities in clinical settings.

To the best of our knowledge, this is the first study to demonstrate that (1) cTBS modulates both regional GABA concentrations and cortical activity in human higher cognition - semantic memory, and that (2) changes in GABA levels are closely linked to changes in regional activity induced by cTBS. These results suggest that GABAergic activity may be the mechanism by which cTBS induces long-lasting after-effects on cortical excitability, leading to behavioural changes. Previous studies in animals and humans have also suggested that cTBS can induce LTD-like effects on cortical synapses and is associated with the GABAergic system in the cortex (*Huang et al., 2011*; *Funke and Benali, 2011*; *Huang et al., 2007*; *Lenz et al., 2016*; *Romero et al., 2022*; *Trippe et al., 2009*). Another study employing MRS found that cTBS increased regional GABA concentrations at the primary motor cortex in healthy subjects (*Stagg et al., 2009*). These findings suggest that cTBS activates a population of cortical GABAergic interneurons, leading to the increase in GABAergic activity (*Di Lazzaro et al., 2005*; *Ziemann et al., 1996*). As a major inhibitory neurotransmitter, GABA has been shown to have a negative correlation with BOLD signal changes (*Duncan et al., 2014*; *Northoff et al., 2007*). Previously, we demonstrated this negative relationship between ATL GABA levels and BOLD signal changes in the ATL during semantic processing (*Jung et al., 2017*), indicating a potential role of GABA in shaping the functions/computations of the cortex. Here, we further demonstrated that the increase in GABA induced by cTBS was negatively correlated with the reduction of BOLD signal responses in the ATL following cTBS, during semantic processing. Our findings suggest a crucial role for GABAergic inhibition in the ATL shaping the local neural functioning underpinning semantic memory and its neuroplasticity. The GABAergic inhibition confines the propagation of excitatory signalling, thereby maintaining the functional organisation of the cortex (*Tremblay et al., 2016*), and the modulation of cortical GABAergic inhibition drives experience-dependent plasticity in cognition (*Schmidt-Wilcke et al., 2018*; *Boroojerdi et al., 2001*).

GABA exists in two distinct neuronal pools: cytoplasmic GABA, which is involved in metabolism, and vesicular GABA, which plays a role in inhibitory synaptic neurotransmission (*Martin and Rimvall, 1993*). In addition to intracellular GABA, extracellular GABA exerts tonic inhibition through extra-synaptic $GABA_A$ receptors (*Semyanov et al., 2004*). MRS is capable of detecting the total concentration of GABA in the voxel of interest, but it cannot differentiate between different pools of GABA (*Stagg et al., 2011b*; *Puts and Edden, 2012*). Some studies have suggested that MRS-measured GABA signals reflect GABAergic tonic inhibition rather than synaptic GABA signalling (*Stagg et al., 2011c*; *Mooney et al., 2017*), whereas other studies have failed to replicate this relationship (*Dyke et al., 2017*; *Hermans et al., 2018*). A recent study has shown a link between MRS-measured GABA and phasic synaptic GABAergic activity (*Lea-Carnall et al., 2023*). Although findings of previous studies have been mixed, changes in GABA levels observed in this study may reflect cTBS-modulated GABAergic neurotransmission, which encompasses both tonic and synaptic GABAergic activity. This GABAergic activity shapes the selective response profiles of neurons in the cortex (*Isaacson and Scanziani, 2011*).

Inverted U-shaped models have been previously considered in the field of neuroscience, specifically in terms of the relationship between the concentration of neurotransmitters such as dopamine, acetylcholine and noradrenaline, and the level of neural activity (*Aston-Jones and Cohen, 2005*; *Cools and D'Esposito, 2011*; *Vijayraghavan et al., 2007*; *He and Zempel, 2013*). Recent studies suggest that this relationship also applies to behaviour, where moderate levels of neural activity are linked to optimal performance (for a review, see *Northoff and Tumati, 2019*). For example, *Ferri et al., 2017* showed an inverted U-shaped relationship between excitation and inhibition balance and multisensory integration, where extreme values impair functionality while intermediate values enhance it, even in healthy individuals. Our findings revealed a non-linear relationship between GABA levels in the ATL and semantic function, indicating that individual variations in semantic task performance can be explained by an inverted-U-shape pattern (*Figure 5A*). Specifically, for relatively greater levels of GABA in the ATL, with lower task-induced regional activity, were associated with better semantic processing in healthy participants (*Jung et al., 2017*). That is, individuals with better semantic memory abilities show more specific cortical activity in the ATL, which is linked to higher concentrations of inhibitory GABA. Extreme levels of GABA can be found in studies with dementia patients and pharmacological studies with GABA agonists. Recent studies have reported decreased GABA levels in Alzheimer's disease (*Xu et al., 2020*; *Jiménez-Balado and Eich, 2021*) and fronto-temporal dementia (*Murley et al., 2020*; *Perry et al., 2022*) in relation to their cognitive impairments such as memory and language. In fact, GABA agonists like midazolam have been found to improve verbal generation in anxiety patients by increasing GABAergic function (*Snyder et al., 2010*). On the other hand, healthy participants who received GABA supplementation (such as baclofen) have been found to have decreased task performance (*Lim and Aquili, 2021*). Overall, optimal, elevated levels of GABA in the ATL may aid in refining stimulated widespread semantic representations through local inhibitory processes. However, our findings should be interpreted with caution due to the limitation of having fewer data points in the latter half (right side) of the inverted U-shaped curve. Future studies incorporating GABA agonists could help further validate and refine these findings.

This inverted U-shaped model could also explain inter-individual variability in cTBS-induced neuroplasticity in the ATL in semantic processing. Our data demonstrated that cTBS over ATL increased regional GABA concentrations, but there was inter-individual variability in GABA level changes in response to cTBS (*Figure 2*). Our previous investigation (*Jung et al., 2022*) showed that the pre-interventional neurochemical state was crucial in predicting cTBS-induced changes in semantic memory. Specifically, cTBS over the ATL inhibited the semantic task performance (i.e. reduced accuracy) of individuals with initially higher concentration of GABA in the ATL, linked to better semantic capacity. However, cTBS had a null or even facilitatory effect on individuals with lower semantic ability with relatively lower GABA levels in the ATL. This study suggests that individuals with higher GABA levels in the ATL were more likely to respond to cTBS, exhibiting inhibitory effects on semantic task performance (responders), while individuals with lower GABA concentrations and lower semantic ability were less likely to respond or even showed facilitatory effects after ATL cTBS (non-responders). The current study revealed a non-linear, inverted-U-shape relationship between GABA levels in the ATL and semantic function, by explaining individual differences in semantic task performance and cTBS responsiveness. As regional GABA increases after cTBS, responders with the optimal level of

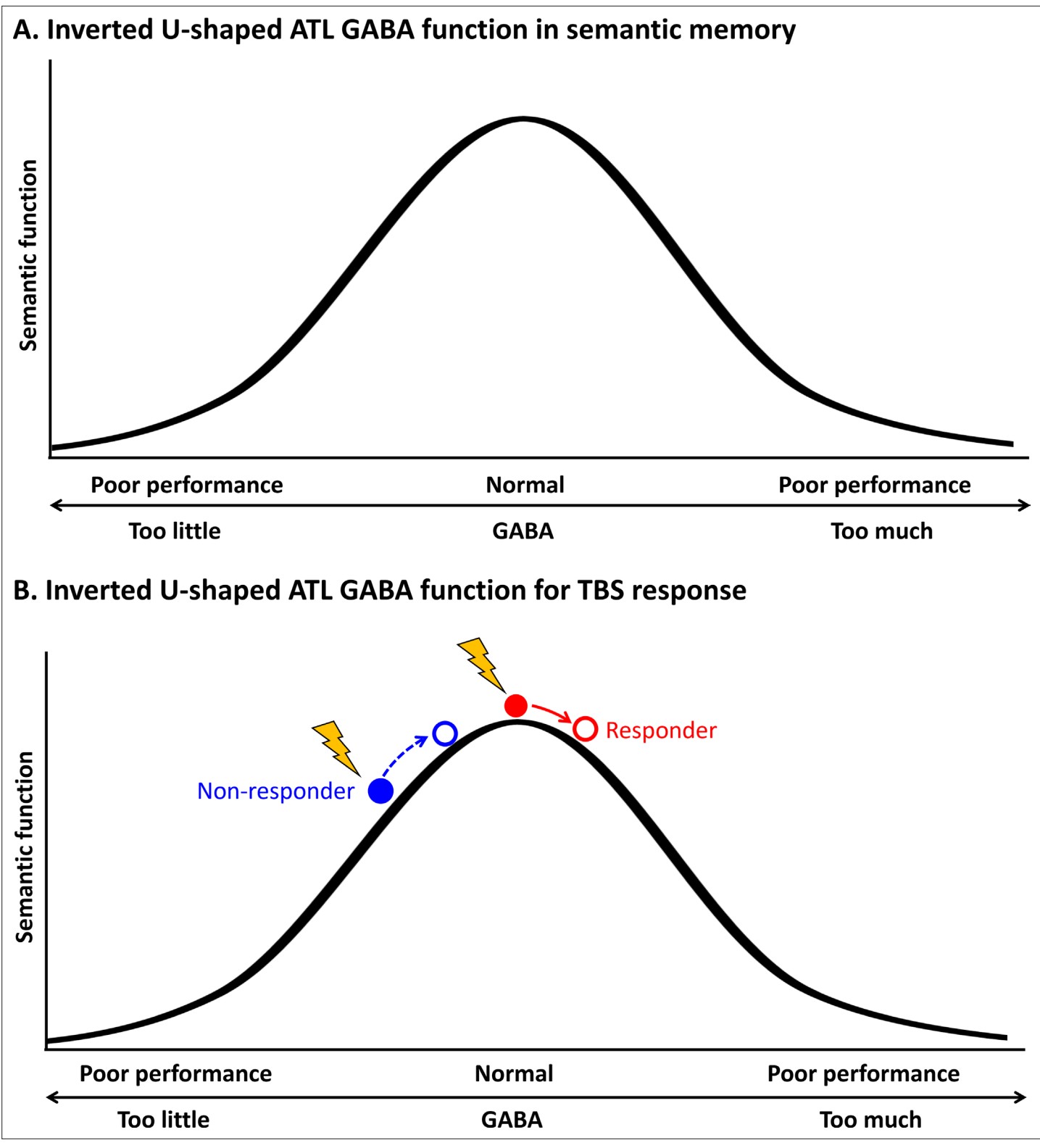

**Figure 5.** Schematic diagram of the relationship between the concentration of GABA in the ATL and semantic function. (**A**) Inverted U-shaped ATL GABA function in semantic memory (**B**) Inverted U-shaped ATL GABA function for cTBS response on semantic memory.

GABA in the ATL would show poorer semantic performance, whereas non-responders could exhibit no changes or even better semantic performance with GABA increase (*Figure 5B*). This relationship is similar to the inverted U-shaped relationship between dopamine action in the prefrontal cortex (PFC) and cognitive control, whereby moderate levels of dopamine lead to optimal cognitive performance (*Cools et al., 2019*). The effects of dopaminergic drugs on PFC function also depend on baseline levels of working memory performance (for a review, see *Cools and D'Esposito, 2011*), explaining the effects of dopaminergic drugs on cognitive performance in individuals with varying working memory capacities (*Kimberg et al., 1997*; *Gibbs and D'esposito, 2005*; *Frank and O'Reilly, 2006*).

Our findings provide novel evidence of a direct link from neurochemical modulations to cortical responses in the brain, highlighting substantial individual variability in semantic memory and plasticity. In addition, the current study represents an important replication and extension of previous findings regarding the role of GABAergic inhibition in semantic memory. These results offer fundamental insights into the mechanisms underlying the maintenance and alteration of functional cortical organisation in response to perturbations. Our study has important implications for the development of personalised therapeutic interventions aimed at modulating neurochemical systems to restore or enhance higher cognitive function in humans.

# Materials and methods

## Key resources table

| Reagent type (species) or resource | Designation | Source or reference | Identifiers | Additional information |
|---|---|---|---|---|
| Software, algorithm | SimNIBS 3.2 | SimNIBS | n/a | |
| Software, algorithm | SPM8 | Statistical Parametric Mapping | RRID:SCR_007037 | |
| Software, algorithm | jMRUI5.1 | jMRUI | RRID:SCR_021893 | |
| Software, algorithm | MATLAB | MathWorks | RRID:SCR_001622 | |
| Software, algorithm | SPSS, Version 25 | IBM | RRID:SCR_002865 | |
| Software, algorithm | RStudio | RStudio | RRID:SCR_000432 | |
| Other | Magstim Super Rapid stimulator | MagStim | n/a | Transcranial Magnetic stimulation |
| Other | 3T Philips Achieva MRI | Philips | n/a | Magnetic Resonance Imaging Scanner |

## Participants

Nineteen healthy English native speakers (9 females, mean age = 25.9 ± 5.8 years, age range: 19–38) participated in this study. The sample size was calculated based on a previous study (*Jung and Lambon Ralph, 2016*), which indicated that to achieve $\alpha$=0.05, power = 80% for the critical interaction between TMS and task, then N≥17 was required. A participant completed one session (ATL stimulation) only. All participants were right-handed (*Oldfield, 1971*). All participants provided written informed consent to participate in the study and to publish the results. The study was conducted at the University of Manchester and approved by the ethics committee of the University of Manchester (REC ref:04/Q1405/66).

To explore the role of GABA in semantic memory function, we used the data previously published (*Jung et al., 2017*). Data from twenty healthy, right-handed native English speakers were included (7 males, mean age = 23 ± 4 years, age range: 20–36).

## Experimental design and procedure

Participants were asked to visit two times for the study. In each visit, the target region was identified prior to the baseline scan. Participants had multimodal imaging (MRS and fMRI). Then, participants were removed from the scanner and cTBS was performed in a separate room. Following cTBS, participants were repositioned into the scanner and had the second multimodal imaging (*Figure 1A*).

We used the same paradigm for the multimodal imaging from our previous study (*Jung et al., 2017*). During MRS, participants were asked to be relaxed with eyes open. Participants performed

a semantic association decision task and pattern matching as a control task during fMRI scanning (*Figure 1B*). The semantic association decision task required a participant to choose which of two pictures at the bottom of the screen was more related in meaning to a probe picture presented on the top of the screen. The items for the semantic association task were from the Pyramids and Palm Tree test (*Howard and Patterson, 1992*) and Camel and Cactus test (*Bozeat et al., 2000*). Items for the pattern matching task were created by scrambling the pictures used in the semantic association task. In the pattern matching task, a participant was asked to identify which of two patterns at the bottom was visually identical to a probe pattern on the top (*Figure 1B*). Participants were required to press one of two buttons designating two choices in a trial. In each trial, there was a fixation for 500ms followed by the stimuli for 4500ms. A task block had four trials of each task. There were 9 blocks of each task interleaved (e.g. A-B-A-B) with a fixation for 4000ms during fMRI. Total scanning time was about 8 min. E-prime software (Psychology Software Tools Inc, Pittsburgh, USA) was used to display stimuli and to record responses.

## Transcranial magnetic stimulation

A Magstim Super Rapid stimulator (MagStim Company, Whitland, UK) with a Figure of eight coil (70 mm standard coil) was used to deliver cTBS over the left ATL or vertex with a week gap between the stimulation (*Figure 1D*). cTBS consisted of bursts containing 3 pulses at 50 Hz (*Huang et al., 2005*) and was applied at 80% of the resting motor threshold (RMT), which previously showed inhibitory effects on semantic processing in the ATL (*Jung and Lambon Ralph, 2016*). RMT was established for each individual, defined as the minimum intensity of stimulation required to produce twitches on 5 of 10 trials from the right first dorsal interosseous (FDI) muscle when the participant was at rest. The average stimulation intensity (80% RMT) was 49.2% ranging from 38% to 60%.

Previous rTMS studies targeted a lateral ATL site 10 mm posterior to the temporal pole on the middle temporal gyrus (MTG; *Pobric et al., 2010*; *Lambon Ralph et al., 2009*; *Pobric et al., 2009*), aligning with the broader ATL region typically atrophied in semantic dementia (*Binney et al., 2010*). However, distortion-corrected fMRI studies (*Visser et al., 2012*; *Binney et al., 2010*) have revealed graded activation differences across the ATL, with peak activation in the ventromedial ATL. Based on these findings, we selected the target site in the left ATL (MNI –36 –15 –30) from a prior distortion-corrected fMRI study (*Visser et al., 2012*; *Binney et al., 2010*) that employed the same tasks as our study (for further details, see the Supplementary Information). This coordinate was transformed to each individual's native space using Statistical Parametric Mapping software (SPM8, Wellcome Trust Centre for Neuroimaging, London, UK). T1 images were normalised to the MNI template and then the resulting transformations were inverted to convert the target MNI coordinate back to the individual's untransformed native space coordinate. These native-space ATL coordinates were subsequently utilised for frameless stereotaxy, employing the Brainsight TMS-MRI co-registration system (Rogue Research, Montreal, Canada). The vertex (Cz) was designated as a control site following the international 10–20 system.

SimNIBS 3.2 (*Thielscher et al., 2015*) was used to calculate the individual electric field of cTBS. The pipeline by *Nielsen et al., 2018* was utilized to generate the individual head model consisting of five tissue types: grey matter (GM), white matter (WM), cerebrospinal fluid (CSF), skull, and scalp. Then, the fixed conductivity values implemented in SimNIBS were applied for each tissue type. The electric field interpolation was performed using *Saturnino et al., 2019* and computed the electrical field at the centre of GM in the ATL and vertex. Finally, we averaged the individual electrical field for the ATL (*Figure 2C*) and vertex (*Figure 2—figure supplement 4*).

## Magnetic resonance imaging acquisition

A 3T Philips Achieva MRI scanner was used to acquire data with a 32-channel head coil with a SENSE factor of 2.5. Structural images were acquired using a magnetisation prepared rapid acquisition gradient echo (MPRAGE) sequence (TR = 8.4 ms, TE = 3.9 ms, slice thickness 0.9 mm, in-plane resolution 0.94 × 0.94 mm).

MRS data were acquired using the GABA-edited MEGA-PRESS sequence (*Mullins et al., 2014*) (TR = 2000ms, TE = 68ms). The voxel of interest (VOI) was manually placed in the left ventrolateral ATL (voxel size = 40 x 20 x 20 mm), avoiding hippocampus or vertex (voxel size = 30 x 30 x 30 mm), Cz, guided by international 10–20 electrode system (*Klem et al., 1999*; *Figure 1C*). Spectra were

acquired in interleaved blocks of four scans with application of the MEGA inversion pulses at 1.95 ppm to edit GABA signal (100 repeats at the ATL VOI and 75 repeats at the vertex VOI). Measurements from the ATL VOI with current protocol provided a robust measure of GABA and glx concentrations (*Jung et al., 2017*; *Sanaei Nezhad et al., 2018*; *Sanaei Nezhad et al., 2020*). A total of 1024 sample points were collected at a spectral width of 2 kHz.

A dual-echo fMRI protocol developed by *Halai et al., 2014* was employed to maximise signal-to-noise (SNR) in the ATL (TR = 2.8 s, TE = 12ms and 35ms, 42 slices, 96 × 96 matrix, 240 × 240 × 126 mm FOV, slice thickness 3 mm, in-plane resolution 2.5 × 2.5).

## MRS analysis

Java-based magnetic resonance user's interface (jMRUI5.1, EU project http://www.jmrui.eu/) (*Naressi et al., 2001*) was used to analyse MRS data. Raw data were corrected using the unsuppressed water signal from the same VOI, eddy current correction, a zero-order phasing of array coil spectra. Residual water was removed using Hankel-Lanczos singular value decomposition (*Cabanes et al., 2001*). Advanced Magnetic Resonance (AMARES; *Laudadio et al., 2002*) was used to quantify neurochemicals including GABA, glx, and NAA. The exclusion criteria for data were as follows: Cramér-Rao bounds >50%, water linewidths at full width at half maximum (FWHM) >20 Hz, and SNR <40. A subject was discarded from the analysis due to poor quality of MRS. GABA and glx values are reported as a ratio to NAA as we previously reported (*Jung et al., 2017*).

Statistical Parametric Map (SPM8, http://www.fil.ion.ucl.ac.uk/spm/) was used to calculate the contributions of GM and WM to the VOI from the structural image. Then voxel registration was performed using custom-made scripts developed in MATLAB by Dr. Nia Goulden, which can be accessed at http://biu.bangor.ac.uk/projects.php.en. The calculation of tissue types within the VOI provided the percentage of each tissue type. As GABA levels are substantially higher (twofold) in the GM than WM (*Jensen et al., 2005*), we used GM as a covariate in the analysis. There was no significant difference in GM volume before and after the stimulation (ps >0.5) and a significant correlation between GM volumes before and after stimulation in both VOIs (ATL stimulation: $r=0.75$, $p<0.001$ in the ATL, $r=0.67$, $p=0.003$ in the vertex; Vertex stimulation: $r=0.68$, $p=0.008$ in the ATL, $r=0.72$, $p<0.001$). The results of tissue segmentation are summarised in *Supplementary file 1*.

## fMRI analysis

fMRI data were processed using SPM8. Dual gradient echo images were realigned, corrected for slice timing, and averaged using in-house MATLAB code developed by *Halai et al., 2014*. The EPI volumes were coregistered into the structural image, spatially normalised to the MNI template using DARTEL(diffeomorphic anatomical registration through an exponentiated lie algebra) toolbox (*Ashburner, 2007*), and smoothed with an 8 mm full-width half-maximum Gaussian filter.

A general linear model (GLM) was used to perform statistical analyses. A design matrix was modelled with task conditions, semantic, control and baseline for each individual along with six motion parameters as regressors. A contrast of interest (semantic >control) for each participant was calculated. One-sample t-test was performed to estimate the contrast of interest at the group level. Clusters were considered significant when passing a threshold of p FWE-corrected <0.05, with at least 100 contiguous voxels.

Regions-of-interest (ROI) analysis was conducted using Marsbar (*Brett et al., 2002*). The mean signal changes of VOIs were extracted for semantic task condition before and after the stimulation.

A voxel-wise simple regression analysis was conducted to identify the local maxima of voxels within the MRS ATL VOI correlating with its BOLD response with GABA levels in the contrast of interest (semantic >control). Local maxima of correlation were estimated on a voxel level, setting the threshold to $p < 0.05$ FWE after small-volume correction.

## Statistical analysis

For behavioural data, accuracy and reaction time (RT) were calculated for each individual. We computed the inverse efficiency (IE) score (RT/1-the proportion of error) to combine the accuracy and RT and calculated the cTBS effect (POST-PRE session). A 2×2 repeated measures ANOVA was conducted with stimulation (ATL vs. vertex) and task (semantic vs. control) as within-subject factors. *Post hoc* paired t-tests were conducted.

To investigate the individual-level effects of cTBS, participants were categorised based on changes in their semantic task performance following the ATL stimulation. Prior to classification, task performance was adjusted to account for practice effects by normalising ATL stimulation performance relative to vertex stimulation performance. Individuals exhibiting a decline in task performance post-ATL cTBS in comparison to the pre-stimulation session were classified as *responders*, while those showing no change or improvement in performance were categorised as *non-responders*. Subsequently, there were nine responders and eight non-responders. A 2×2 × 2 ANOVA was conducted with task (semantic vs. control) and session (PRE vs. POST) as within-subject factors, and group (responders vs. non-responders) as a between-subject factor was performed. *Post hoc* t-tests were conducted to examine differences in task performance between responders and non-responders. Additionally, a 2x2 ANOVA with session (pre vs. post) as a within-subject factor and with group (responders vs. non-responders) as a between-subject factor was conducted to examine the effects of group in ATL GABA levels. *Post hoc* t-tests were performed to investigate the cTBS responsiveness on semantic task performance and ATL GABA levels.

Partial correlation analysis was performed to illustrate the relationship between the ATL GABA levels and semantic task performance, accounting for GM volume, age and sex.

Regression analyses (linear and quadratic models) were conducted to explore the relationship between the ATL GABA levels and semantic task performance. The individual ATL GABA levels were adjusted by GM volume, age, and sex, using multiple regression analysis. In order to determine the best-fit model, we calculated the Bayesian Information Criterion (BIC) as a measure of model fitness (**Vrieze, 2012**) and performed a partial F-test.

Statistical analyses were undertaken using Statistics Package for the Social Sciences (SPSS, Version 25, IBM Cary, NC, USA) and RStudio (2023).

## Acknowledgements

This research was supported by AMS Springboard (SBF007\100077) to JJ and an Advanced ERC award (GAP: 670428 -BRAIN2MIND_NEUROCOMP), MRC programme grant (MR/R023883/1), and intramural funding (MC_UU_00030/9) to MALR.

## Additional information

### Funding

| Funder | Grant reference number | Author |
|---|---|---|
| Academy of Medical Sciences | SBF007\100077 | JeYoung Jung |
| European Research Council | GAP: 670428 -BRAIN2MIND_ NEUROCOMP | Matthew A Lambon Ralph |
| Medical Research Council | MR/R023883/1 | Matthew A Lambon Ralph |
| Medical Research Council | MC_UU_00030/9 | Matthew A Lambon Ralph |

The funders had no role in study design, data collection and interpretation, or the decision to submit the work for publication.

### Author contributions

JeYoung Jung, Conceptualization, Resources, Data curation, Formal analysis, Funding acquisition, Validation, Investigation, Visualization, Methodology, Writing – original draft, Writing – review and editing; Steve Williams, Methodology, Writing – review and editing; Matthew A Lambon Ralph, Conceptualization, Funding acquisition, Investigation, Writing – original draft, Writing – review and editing

### Author ORCIDs

JeYoung Jung https://orcid.org/0000-0003-3739-7331
Matthew A Lambon Ralph https://orcid.org/0000-0001-5907-2488

### Ethics

All participants provided written informed consent to participate in the study and to publish the results. The study was approved by the ethics committee of the University of Manchester (REC ref:04/Q1405/66).

Reviewer #1 (Public review): https://doi.org/10.7554/eLife.91771.4.sa1
Reviewer #3 (Public review): https://doi.org/10.7554/eLife.91771.4.sa2
Author response https://doi.org/10.7554/eLife.91771.4.sa3

## Additional files

### Supplementary files

Supplementary file 1. Supplementary methods and tables.
MDAR checklist

### Data availability

Data and analysis code are available on OSF (https://doi.org/10.17605/OSF.IO/PMQXH).

The following dataset was generated:

| Author(s) | Year | Dataset title | Dataset URL | Database and Identifier |
|---|---|---|---|---|
| Jung J | 2025 | Data and code: The role of GABA in semantic memory and its neuroplasticity | https://doi.org/10.17605/OSF.IO/PMQXH | Open Science Framework, 10.17605/OSF.IO/PMQXH |

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
