## [Editor Report · eLife Assessment]

Jung et al. present **valuable** work on the relationship between gamma-aminobutyric acid (GABA) levels within the anterior temporal lobes (ATL) to semantic memory while accounting for inter-individual differences. They provide **solid** evidence suggesting that inhibitory continuous theta burst transcranial magnetic stimulation (cTBS TMS) increased GABA concentration and decreased the blood-oxygen dependent signal (BOLD) during a semantic task. The results will be of interest to researchers studying the neurobiology of semantic cognition.

---

## [Referee Report · Reviewer #1 (Public review)]

This study presents valuable findings on the GABA and BOLD changes induced by continuous theta burst stimulation (cTBS) and on the relationships between ATL GABA level and performance in a semantic task. However, I'm afraid that the current results are incomplete to support some primary claims of the paper, for example, the purported inverted-U-shaped relationship between GABA levels in the ATL and semantic task performance. The influence of practice effects also complicates the interpretation of the results. Additional concerns include potential double dipping in the analysis depicted in Figure 3A and the use of inconsistent behavioral measures (IE and accuracy) across various analyses.

The authors have made two beneficial revisions in this round: (1) acknowledging the insufficient data points supporting the inverted U-shaped curve; (2) attempting to control for practice effects. However, I believe unresolved issues remain:

(1) The authors have not addressed my specific concern about Figure 4D - the analysis attempts to fit an inverted U-shaped curve to the data without distinguishing between data points influenced by practice effects and those unaffected, rendering its reliability questionable.

(2) The authors appear to have misunderstood my question regarding Figure 3A. This issue is unrelated to practice effects. My point was that even if we randomly generated pre- and post-test data points and grouped/analyzed them according to the authors' methodology, we would still likely reproduce the pattern in Figure 3A due to the double dipping problem. Thus, this statistical analysis and its conclusions currently lack methodological validity.

(3) Regarding the inconsistency in behavioral measures, the authors' explanation fails to remove my concerns. If the authors argue that accuracy is the most appropriate behavioral dependent variable for this study, why did they employ inverse efficiency in some of their analyses? My understanding is that a study should either consistently use the single most suitable measure or report multiple measures while providing adequate discussion of inconsistent results.

---

## [Referee Report · Reviewer #3 (Public review)]

As a result of a number of rounds of reviews and consultations between reviewers, Jung et al. present important work on the relationship between gamma-aminobutyric acid (GABA) levels within the anterior temporal lobes (ATL) to semantic memory while accounting for inter-individual differences. They provide solid evidence suggesting that inhibitory continuous theta burst transcranial magnetic stimulation (cTBS TMS) increased GABA concentration and decreased the blood-oxygen dependent signal (BOLD) during a semantic task.

The authors fully addressed my comments from the first and second rounds of reviews, and I do not have additional concerns. I have, however, scaled down my short assessment, given the concerns of reviewers 1 and 2.

---

## [Author Response]

The following is the authors’ response to the previous reviews

**Public Reviews:**

**Reviewer #1 (Public review):**
Summary:This study examined the changes in ATL GABA levels induced by cTBS and its relationship with BOLD signal changes and performance in a semantic task. The findings suggest that the increase in ATL GABA levels induced by cTBS is associated with a decrease in BOLD signal. The relationship between ATL GABA levels and semantic task performance is nonlinear, and more specifically, the authors propose that the relationship is an inverted U-shaped relationship.Strengths:The findings of the research regarding the increase of GABA and decrease of BOLD caused by cTBS, as well as the correlation between the two, appear to be reliable. This should be valuable for understanding the biological effects of cTBS.Weakness:I am pleased to see the authors' feedback on my previous questions and suggestions, and I believe the additional data analysis they have added is helpful. Here are my reserved concerns and newly discovered issues.(1) Regarding the Inverted U-Shaped Curve In the revised manuscript, the authors have accepted some of my suggestions and conducted further analysis, which is now presented in Figure 3B. These results provide partial support for the authors' hypothesis. However, I still believe that the data from this study hardly convincingly support an inverted U-shaped distribution relationship.The authors stated in their response, "it is challenging to determine the optimal level of ATL GABA," but I think this is achievable. From Figures 4C and 4D, the ATL GABA levels corresponding to the peak of the inverted U-shaped curve fall between 85 and 90. In my understanding, this can be considered as the optimal level of ATL GABA estimated based on the existing data and the inverted U-shaped curve relationship. However, in the latter half of the inverted U-shaped curve, there are quite few data points, and such a small number of data points hardly provides reliable support for the quantitative relationship in the latter half of the curve. I suggest that the authors should at least explicitly acknowledge this and be cautious in drawing conclusions. I also suggest that the authors consider fitting the data with more types of non-linear relationships, such as a ceiling effect (a combination of a slope and a horizontal line), or a logarithmic curve.

We appreciate R1’s comments. Inverted U-shaped relationships are well-established in neuroscience, particularly in the context of neurotransmitter concentrations (e.g., dopamine, acetylcholine, noradrenaline) and their influence on cognitive functions such as working memory and cognitive control (Aston-Jones & Cohen., 2005; Cools & D'Esposito., 2011; Vijayraghavan et al., 2007; He & Zempel., 2013). Recently, Ferri et al. (2017) demonstrated an inverted U-shaped relationship between excitation-inhibition balance (EIB: the ratio of Glx and GABA) and multisensory integration, showing that both excessive and insufficient inhibition negatively impact functionality. Given that GABA is the brain’s primary inhibitory neurotransmitter, our findings suggest that ATL GABA may play a similar regulatory role in semantic memory function.

While our statistical modelling approach demonstrated that the inverted U-shaped function was the best-fitting model for our current data in explaining the relationship between ATL GABA and semantic memory, we acknowledge the limitation of having fewer data points in the latter half (right side) of the curve, where excessive ATL GABA levels are associated with poorer semantic performance. Following R1’s suggestion, we have explicitly acknowledged this limitation in the revised manuscript and exercised caution in our discussion.

Discussion, p.17, line 408

"However, our findings should be interpreted with caution due to the limitation of having fewer data points in the latter half (right side) of the inverted U-shaped curve. Future studies incorporating GABA agonists could help further validate and refine these findings".

Following R1’s latter suggestion, we tested a logarithmic curve model. The results showed significant relationships between ATL GABA and semantic performance (R^2^ = 0.544, p < 0.001) and between cTBS-induced changes in ATL GABA and semantic performance (R^2^ = 0.202, p < 0.001). However, the quadratic (inverted U-shaped) model explained more variance than the logarithmic model, as indicated by a higher R^2^ and lower BIC. Model comparisons further confirmed that the inverted U-shaped model provided the best fit for both ATL GABA in relation to semantic performance (Fig. 4C) and cTBS-induced ATL GABA changes in relation to semantic function (Fig. 4D).

**Author response table 1. sa3table1:** 

Model	ATL GABA and semantic performance		cTBS-induced changes in ATL GABA and semantic performance	
	R^(2)	BIC	R^(2)	BIC
Logarithmic curve	0.54	245	0.20	229
Quadratic curve (inverted U shaped)	0.67	233	0.33	227
Model comparison (F-test)	F=14.01,p=0.00067		F=6.24,p=0.018	

(2) In Figure 2F, the authors demonstrated a strong practice effect in this study, which to some extent offsets the decrease in behavioral performance caused by cTBS. Therefore, I recommend that the authors give sufficient consideration to the practice effect in the data analysis.One issue is the impact of the practice effect on the classification of responders and non-responders. Currently, most participants are classified as non-responders, suggesting that the majority of the population may not respond to the cTBS used in this study. This greatly challenges the generalizability of the experimental conclusions. However, the emergence of so many non-responders is likely due to the prominent practice effect, which offsets part of the experimental effect. If the practice effect is excluded, the number of responders may increase. The authors might estimate the practice effect based on the vertex simulation condition and reclassify participants after excluding the influence of the practice effect.Another issue is that considering the significant practice effect, the analysis in Figure 4D, which mixes pre- and post-test data, may not be reliable.

We appreciate Reviewer 1’s thoughtful comments regarding the practice effect and its potential impact on our findings. Our previous analysis revealed a strong practice effect on reaction time (RT), with participants performing tasks faster in the POST session, regardless of task condition (Fig. S3). Given our hypothesis that inhibitory ATL cTBS would disrupt semantic task performance, we accounted for this by using inverse efficiency (IE), which combines accuracy and RT. This analysis demonstrated that ATL cTBS disrupted semantic task performance compared to both control stimulation (vertex) and control tasks, despite the practice effect (i.e., faster RT in the POST session), thereby supporting our hypothesis. These findings may suggest that the effects of ATL cTBS were more subtly reflected in semantic task accuracy rather than RT.

Regarding inter-individual variability in response to rTMS/TBS, prior studies have shown that 50–70% of participants are non-responders, either do not respond or respond in an unexpected manner (Goldsworthy et al., 2014; Hamada et al., 2013; Hinder et al., 2014; Lopez-Alonso et al., 2014; Maeda et al., 2000a; Müller-Dahlhaus et al., 2008). Our previous study (Jung et al., 2022) using the same semantic task and cTBS protocol was the first to explore TBS-responsiveness variability in semantic memory, where 12 out of 20 participants (60%) were classified as responders. The proportion of responders and non-responders in the current study aligns with previous findings, suggesting that this variability is expected in TBS research.

However, we acknowledge R1’s concern that the strong practice effect may have influenced responder classification. To address this, we estimated the practice effect using the vertex stimulation condition and reclassified participants accordingly by adjusting ATL stimulation performance (IE) relative to vertex stimulation performance (IE). This reclassification identified nine responders (an increase of two), aligning with the typical responder proportion (52%) reported in the TBS literature. Overall, we replicated the previous findings with improved statistical robustness.

A 2×2×2 ANOVA was conducted with task (semantic vs. control) and session (PRE vs. POST) as within-subject factors, and group (responders vs. non-responders) as a between-subject factor. The analysis revealed a significant interaction between the session and group (F_1, 15_ = 10.367, p = 0.006), a marginally significant interaction between the session and task (F_1, 15_ = 4.370, p = 0.054), and a significant 3-way interaction between the session, task, and group (F_1, 15_ = 7.580, p = 0.015). *Post hoc* t-tests showed a significant group difference in semantic task performance following ATL stimulation (t = 2.349, p = 0.033). *Post hoc* paired t-test demonstrated that responders exhibited poorer semantic task performance following the ATL cTBS (t = -5.281, p < 0.001), whereas non-responders showed a significant improvement (t = 3.206, p = 0.007) (see Figure. 3A).

Notably, no differences were observed between responders and non-responders in the control task performance across pre- and post-stimulation sessions, confirming that the practice effect was successfully controlled (Figure. 3B).

We performed a 2 x 2 ANOVA with session (pre vs. post) as a within subject factor and with group (responders vs. non-responders) as a between subject factor to examine the effects of group in ATL GABA levels. The results revealed a significant main effect of session (F_1, 14_ = 39.906, p < 0.001) and group (F_1, 14_ = 9.677, p = 0.008). *Post hoc* paired t-tests on ATL GABA levels showed a significant increase in regional ATL GABA levels following ATL stimulation for both responders (*t* = -3.885, *p* = 0.002) and non-responders (*t* = -4.831, *p* = 0.001). Furthermore, we replicated our previous finding that baseline GABA levels were significantly higher in responders compared to non-responders (t = 2.816, p = 0.007) (Figure. 3C). This pattern persisted in the post-stimulation session (t = 2.555, p = 0.011) (Figure. 3C).

Accordingly, we have revised the Methods and Materials (p 26, line 619), Results (p11, line 233-261), and Figure 3.

(3) The analysis in Figure 3A has a double dipping issue. Suppose we generate 100 pairs of random numbers as pre- and post-test scores, and then group the data based on whether the scores decrease or increase; the pre-test scores of the group with decreased scores will have a very high probability of being higher than those of the group with increased scores. Therefore, the findings in Figure 3A seem to be meaningless.

Yes, we agreed with R1’s comments. However, Figure 3A illustrates interindividual responsiveness patterns, while Figure 3B demonstrates that these results account for practice effects, incorporating new analyses.

(4) The authors use IE as a behavioral measure in some analyses and use accuracy in others. I recommend that the authors adopt a consistent behavioral measure.

We appreciate Reviewer 1’s suggestion. In examining the relationship between ATL GABA and semantic task performance, we have found that only semantic accuracy—not reaction time (RT) or inverse efficiency (IE)—shows a significant positive correlation and regression with ATL GABA levels and semantic task-induced ATL activation, both in our previous study (Jung et al., 2017) and in the current study. ATL GABA levels were not correlated with semantic RT (Jung et al., 2017: r = 0.34, p = 0.14, current study: r = 0.26, p = 0.14). It should be noted that there were no significant correlations between ATL GABA levels and semantic inverse efficiency (IE) in both studies (Jung et al., 2017: r = 0.13, p = 0.62, current study: r = 0.22, p = 0.44). As a result, we found no significant linear and non-linear relationship between ATL GABA levels and RT (linear function R^2^ = 0.21, p = 0.45, quadratic function: R^2^ = 0.17, p = 0.21) and between ATL GABA levels and IE (linear function R^2^ = 0.24, p = 0.07, quadratic function: R^2^ = 2.24, p = 0.12).

The absence of a meaningful relationship between ATL GABA and semantic RT or IE may be due to the following reasons: (1) RT is primarily associated with premotor and motor activation during semantic processing rather than ATL activation; (2) ATL GABA is likely to play a key role in refining distributed semantic representations through lateral inhibition, which sharpens the activated representation (Jung et al., 2017; Liu et al. 2011; Isaacson & Scanziani., 2011). This sharpening process may contribute to more accurate semantic performance (Jung et al., 2017). In our semantic task, for example, when encountering a camel (Fig. 1B), multiple semantic features (e.g., animal, brown, desert, sand, etc.) are activated. To correctly identify the most relevant concept (cactus), irrelevant associations (tree) must be suppressed—a process that likely relies on inhibitory mechanisms. Given this theoretical framework, we have used accuracy as the primary measure of semantic performance to elucidate the ATL GABA function.

**Reviewer #2 (Public review):**
Summary:The authors combined inhibitory neurostimulation (continuous theta-burst stimulation, cTBS) with subsequent MRI measurements to investigate the impact of inhibition of the left anterior temporal lobe (ATL) on task-related activity and performance during a semantic task and link stimulation-induced changes to the neurochemical level by including MR spectroscopy (MRS). cTBS effects in the ATL were compared with a control site in the vertex. The authors found that relative to stimulation of the vertex, cTBS significantly increased the local GABA concentration in the ATL. cTBS also decreased task-related semantic activity in the ATL and potentially delayed semantic task performance by hindering a practice effect from pre to post. Finally, pooled data with their previous MRS study suggest an inverted u-shape between GABA concentration and behavioral performance. These results help to better understand the neuromodulatory effects of non-invasive brain stimulation on task performance.Strengths:Multimodal assessment of neurostimulation effects on the behavioral, neurochemical, and neural levels. In particular, the link between GABA modulation and behavior is timely and potentially interesting.Weaknesses:The analyses are not sound. Some of the effects are very weak and not all conclusions are supported by the data since some of the comparisons are not justified. There is some redundancy with a previous paper by the same authors, so the novelty and contribution to the field are overall limited. A network approach might help here.
**Reviewer #3 (Public review):**
Summary:The authors used cTBS TMS, magnetic resonance spectroscopy (MRS), and functional magnetic resonance imaging (fMRI) as the main methods of investigation. Their data show that cTBS modulates GABA concentration and task-dependent BOLD in the ATL, whereby greater GABA increase following ATL cTBS showed greater reductions in BOLD changes in ATL. This effect was also reflected in the performance of the behavioural task response times, which did not subsume to practice effects after AL cTBS as opposed to the associated control site and control task. This is in line with their first hypothesis. The data further indicates that regional GABA concentrations in the ATL play a crucial role in semantic memory because individuals with higher (but not excessive) GABA concentrations in the ATLs performed better on the semantic task. This is in line with their second prediction. Finally, the authors conducted additional analyses to explore the mechanistic link between ATL inhibitory GABAergic action and semantic task performance. They show that this link is best captured by an inverted U-shaped function as a result of a quadratic linear regression model. Fitting this model to their data indicates that increasing GABA levels led to better task performance as long as they were not excessively low or excessively high. This was first tested as a relationship between GABA levels in the ATL and semantic task performance; then the same analyses were performed on the pre and post-cTBS TMS stimulation data, showing the same pattern. These results are in line with the conclusions of the authors.Comments on revisions:The authors have comprehensively addressed my comments from the first round of review, and I consider most of their answers and the steps they have taken satisfactorily. Their insights prompted me to reflect further on my own knowledge and thinking regarding the ATL function.I do, however, have an additional and hopefully constructive comment regarding the point made about the study focusing on the left instead of bilateral ATL. I appreciate the methodological complexities and the pragmatic reasons underlying this decision. Nevertheless, briefly incorporating the justification for this decision into the manuscript would have been beneficial for clarity and completeness. The presented argument follows an interesting logic; however, despite strong previous evidence supporting it, the approach remains based on an assumption. Given that the authors now provide the group-level fMRI results captured more comprehensively in Supplementary Figure 2, where the bilateral pattern of fMRI activation can be observed in the current data, the authors could have strengthened their argument by asserting that the activation related to the given semantic association task in this data was bilateral. This would imply that the TMS effects and associated changes in GABA should be similar for both sites. Furthermore, it is worth noting the approach taken by Pobric et al. (2007, PNAS), who stimulated a site located 10 mm posterior to the tip of the left temporal pole along the middle temporal gyrus (MTG) and not the bilateral ATL.

We appreciate the reviewer’s constructive comment regarding the focus on the left ATL rather than bilateral ATL in our study. Accordingly, we have added the following paragraph in the Supplementary Information.

“Justification of target site selection and cTBS effects

Evidence suggests that bilateral ATL systems contribute to semantic representation (for a review, see Lambon Ralph., 2017). Consistent with this, our semantic task induced bilateral ATL activation (Fig. S2). Thus, stimulating both left and right ATL could provide a more comprehensive understanding of cTBS effects and its GABAergic function.

Previous rTMS studies have applied inhibitory stimulation to the left vs. right ATL, demonstrating that stimulation at either site significantly disrupted semantic task performance (Pobric et al., 2007, PNAS; Pobric et al., 2010, Neuropsychologia; Lambon Ralph et al., 2009, Cerebral Cortex). Importantly, these studies reported no significant difference in rTMS effects between left and right ATL stimulation, suggesting that stimulating either hemisphere produces comparable effects on semantic processing. In the current study, we combined cTBS with multimodal imaging to investigate its effects on the ATL. Given our study design constraints (including the need for a control site, control task, and control stimulation) and limitations in scanning time, we selected the left ATL as the target region. This choice also aligned with the MRS voxel placement used in our previous study (Jung et al., 2017), allowing us to combine datasets and further investigate GABAergic function in the ATL. Accordingly, cTBS was applied to the peak coordinate of the left ventromedial ATL (MNI -36, -15, -30) as identified by previous fMRI studies (Binney et al., 2010; Visser et al., 2012).

Given that TMS pulses typically penetrate 2–4 cm, we acknowledge the challenge of reaching deeper ventromedial ATL regions. However, our findings indicate that cTBS effectively modulated ATL function, as evidenced by reduced task-induced regional activity, increased ATL GABA concentrations, and poorer semantic performance, confirming that TMS pulses successfully influenced the target region. To further validate these effects, we conducted an ROI analysis centred on the ventromedial ATL (MNI -36, -15, -30), which revealed a significant reduction in ATL activity during semantic processing following ATL stimulation (*t* = -2.43, p = 0.014) (Fig. S7). This confirms that cTBS successfully modulated ATL activity at the intended target coordinate.”

We appreciate R3's comment regarding the approach taken by Pobric et al. (2007, PNAS), who stimulated a site 10 mm posterior to the tip of the left temporal pole along the middle temporal gyrus (MTG). This approach has been explicitly discussed in our previous papers and reviews (e.g., Lambon Ralph, 2014, Proc. Royal Society B). Our earlier use of lateral ATL stimulation at this location (Pobric et al. 2007; Lambon Ralph et al. 2009; Pobric et al. 2010) was based on its alignment with the broader ATL region commonly atrophied in semantic dementia (cf. Binney et al., 2010 for a direct comparison of SD atrophy, fMRI data and the TMS region). Since these original ATL TMS investigations, a series of distortion-corrected or distortion-avoiding fMRI studies (e.g., Binney et al 2010; Visser et al, various, Hoffman et al., various; Jackson et al., 2015) have demonstrated graded activation differences across the ATL. While weaker activation is present at the original lateral ATL (MTG) stimulation site, the peak activation is maximal in the ventromedial ATL—a finding that was also observed in the current study. Accordingly, we selected the ventromedial ATL as our target site for stimulation.

Following these points, we have revised the manuscript in the Methods and Materials.

Transcranial magnetic stimulation p23, line 525-532,

“Previous rTMS studies targeted a lateral ATL site 10 mm posterior to the temporal pole on the middle temporal gyrus (MTG) (Pobric et al. 2007; Lambon Ralph et al. 2009; Pobric et al. 2010), aligning with the broader ATL region typically atrophied in semantic dementia (Binney et al. 2010). However, distortion-corrected fMRI studies (Binney et al. 2010; Visser et al. 2012) have revealed graded activation differences across the ATL, with peak activation in the ventromedial ATL. Based on these findings, we selected the target site in the left ATL (MNI -36, -15, -30) from a prior distortion-corrected fMRI study Binney et al. 2010; Visser et al. 2012 that employed the same tasks as our study (for further details, see the Supplementary Information).”

**Recommendations for the authors:**

**Reviewer #2 (Recommendations for the authors):**
The authors have responded to all my comments and I found most of the responses reasonable and sufficient. However, I have one remaining point: I pointed out before that the scope of this paper is somehow narrow and asked for a network analysis. I found the response to my question somehow puzzling since the authors write:"However, it is important to note that we did not find any significant correlations between ATL GABA changes and cTBS-induced changes in the functional connectivity. Consequently, we are currently preparing another paper that specifically addresses the network-level changes induced by ATL cTBS."I don't understand the logic here. Even in the absence of significant correlations between ATL GABA changes and cTBS-induced changes in connectivity, it would be interesting to know how baseline connectivity is correlated with the induced changes. I am not sure if it is adequate to squeeze another paper out of the dataset instead of reporting it here as suggested.

We apologise that our previous response was not clear. To examine cTBS-induced network-level changes, we conducted ROI analyses targeting key semantic regions, including the bilateral ATL, inferior frontal gyrus (IFG), and posterior middle temporal gyrus (pMTG), as well as Psychophysiological Interactions (PPI) using the left ATL as a seed region. The ROI analysis revealed that ATL stimulation significantly decreased task-induced activity in the left ATL (target region) while increasing activity in the right ATL and left IFG. PPI analyses showed that ATL stimulation enhanced connectivity between the left ATL and the right ATL (both ventromedial and lateral ATL), bilateral IFG, and bilateral pMTG, suggesting that ATL stimulation modulates a bilateral semantic network.

Building on these findings, we conducted Dynamic Causal Modeling (DCM) to estimate and infer interactions among predefined brain regions across different experimental conditions (Friston et al., 2003). The bilateral ventromedial ATL, lateral ATL, IFG, and pMTG were defined as network nodes with mutual connections. Our model examined cTBS effects at the left ATL under both baseline (intrinsic) and semantic task (modulatory) conditions, estimating 56 intrinsic parameters for baseline connectivity and testing 16 different modulatory models to assess cTBS-induced connectivity changes during semantic processing. Here, we briefly summarize the key DCM analysis results: (1) ATL cTBS significantly altered effective connectivity between the left and right lateral and ventromedial ATL in both intrinsic and modulatory conditions; (2) cTBS increased modulatory connectivity from the right to the left ATL compared to vertex stimulation.

Given the complexity and depth of these findings, we believe that a dedicated paper focusing on the network-level effects of ATL cTBS is necessary to provide a more comprehensive and detailed analysis, which extends beyond the scope of the current study. It should be noted that no significant relationship was found between ATL GABA levels and ATL connectivity in both PPI and DCM analyses.

**Reviewer #3 (Recommendations for the authors):**
In response to my comment about the ATL activation being rather medial in the fMRI data and my concern about the TMS pulse perhaps not reaching this site, the authors offer an excellent solution to demonstrate TMS effects to such a medial ATL coordinate. I think that the analyses and figures they provide as a response to this comment and a brief explanation of this result should be incorporated into supplementary materials for methodologically oriented readers. Also, perhaps it would be beneficial to discuss that the effect of TMS on vATL remains a matter of further research to see not just if but also how TMS pulse reaches target coordinates, given the problematic anatomical location of the region.

We appreciate R3’s suggestion. Please, see our reply above.